# Narrow Diameter Dental Implants as an Alternative Treatment for Atrophic Alveolar Ridges. Systematic Review and Meta-Analysis

**DOI:** 10.3390/ma14123234

**Published:** 2021-06-11

**Authors:** Georgina González-Valls, Elisabet Roca-Millan, Juan Manuel Céspedes-Sánchez, Beatriz González-Navarro, Aina Torrejon-Moya, José López-López

**Affiliations:** 1Faculty of Medicine and Health Sciences, School of Dentistry, University of Barcelona, 08907 Barcelona, Spain; georgina.gonzalezvalls@gmail.com (G.G.-V.); erocamil@gmail.com (E.R.-M.); Juanmacesp88@hotmail.com (J.M.C.-S.); beatrizgonzaleznavarro@gmail.com (B.G.-N.); aina.torrejon@gmail.com (A.T.-M.); 2Department of Odontostomatology, Faculty of Medicine and Health Sciences, School of Dentistry, University of Barcelona//Oral Health and Masticatory System Group-IDIBELL (Bellvitge Biomedical Research Institute), University of Barcelona, 08907 Barcelona, Spain; 3Medical Director and Head of Service of the Surgical Medical Area, Odontological Hospital University of Barcelona, University of Barcelona, 08907 Barcelona, Spain

**Keywords:** diameter, narrow-diameter, dental implants, mini dental implants

## Abstract

To determine the marginal bone loss and the survival, success and failure rates of narrow dental implants, a systematic literature search was carried out in the MEDLINE (Pubmed), Cochrane, Scopus, and Scielo databases for articles published between 2010 and 2021. The exclusion criteria were: systematic reviews, case reports, expert opinions; animal studies; samples of less than 10 subjects; follow-up periods of less than 36 months; smokers of minimum 10 cigarettes/day; and articles about mini-implants for orthodontic anchorage. Meta-analyses were performed to assess marginal bone loss and implant survival, success, and failure rates. Fifteen studies were included: 7 clinical trials, 3 randomized clinical trials, 3 cohort studies, and 2 case series. The total number of subjects was 773, in whom 1245 implants were placed. The survival rate for the narrow diameter implants was 97%, the success rate 96.8%, and the failure rate 3%. Marginal bone loss was 0.821 mm. All these data were evaluated at 36 months. Based on the literature, it can be considered that there is sufficient evidence to consider small diameter implants a predictable treatment option. These show favorable survival and success rates and marginal bone loss. All of them are comparable to those of standard diameter dental implants.

## 1. Introduction

Oral rehabilitation with dental implants provides an increase in oral health and quality of life [1,2]. Such a treatment has shown a success rates of up to 98% at 10 years [3], and excellent treatment predictability, which is reflected in countless clinical studies [3,4,5]. Historically, implants have been used and documented mainly with diameters of 3.75 mm and 4.1 mm, being considered standard diameter dental implants (SDI). The indications for these implants have been numerous and treatment protocols have been established with excellent long-term results [6]. However, a disadvantage of SDIs is that bone availability, both of the alveolar ridge horizontally, as well as between tooth and tooth or tooth and implant, is sometimes insufficient [7].

Alveolar ridge resorption begins immediately after tooth extraction and this process is more intense during the first year, when about 60% of the thickness of the alveolar ridge is resorbed [8,9]. In the mandible, ridge resorption is chronic, progressive and directly linked to the duration of edentulism [10].

In addition to the aforementioned, the reduction in bone width may be due to, or aggravated by, other causes such as trauma, malformation, neoplasms, use of removable prostheses and periodontal disease [11]. This causes challenging limitations for implant placement. In these cases, surgical procedures may be necessary to increase insufficient bone volume [11]. However, these procedures require surgical expertise to prevent possible complications such as postoperative pain, infection, nerve damage, bone fractures, bleeding, wound dehiscence, and implant failure. This greater morbidity, together with a high economic cost and a longer surgical and healing time, raises the need for other therapeutic options [11].

Likewise, it is considered that in medically compromised or elderly patients, regenerative procedures carry a high risk of complications [12,13]. Therefore, alternative concepts, such as narrow diameter dental implants (NDI) are leading to increased interest at the clinical and scientific levels. As early as 2014, Schiegnitz et al. [12], indicated that avoiding regenerative procedures or other invasive surgical treatments using NDI can expand treatment options, avoid more invasive procedures, reducing patient morbidity and treatment time [12].

The definition of NDI is not conclusive in published studies and there are several classifications according to different authors. This review has been based on that described by Klein et al. in 2014 [7] which distinguishes between three sub-categories: Category 1: <3.0 mm (“mini-implants”); Category 2: 3.0–3.25 mm; Category 3: 3.30–3.50 mm.

The clinical evidence comparing the NDI with SDI creates controversy in the literature; consequently, and based on the above, the objective of this systematic review is to determine the survival, success and failure rates of the NDI placed in narrow bone compared to the SDI placed in normal bone, as well as the peri-implant marginal bone loss.

## 2. Materials and Methods

### 2.1. Reporting Format

This systematic review has been prepared according to the PRISMA (Preferred Reporting Items for Systematic Reviews and Meta-Analysis) criteria [14]. Before the systematic search, a detailed protocol was developed on the methodology to be followed. The protocol was not registered.

### 2.2. PICO (Population, Intervention, Comparison, Outcomes) Question

Population (P): Partially or totally edentulous patients; Intervention (I): Placement of narrow dental implants in the mandible and/or maxilla; Comparison (C): Standard diameter or larger diameter implants; Outcomes (O): Implant success, survival and failure rates and marginal bone loss.

### 2.3. Eligibility Criteria

Inclusion criteria: (i) articles in which the main objective was to treat partially or totally edentulous patients with narrow dental implants were considered; (ii) clinical trials, cohort studies, and case series in healthy humans; (iii) published in the last 10 years in English or Spanish; (iv) minimum 10 patients treated with NDI; (v) With a minimum follow-up period of 36 months after implant placement. Exclusion criteria: (i) systematic reviews, case reports, and expert opinions; (ii) animal studies; (iii) studies in which the sample was less than 10 subjects; (iv) follow-up period of less than 36 months after implant placement; (v) smokers of more than 10 cigarettes/day; (vi) mini-implants for orthodontic anchoring.

### 2.4. Search Strategy and Study Selection

An electronic search was carried out in the MEDLINE/PubMed, Cochrane, Scopus, and Scielo databases for articles published between 2010 and 2020. The last search was carried out on 5 March 2021.

The following keywords were used: small diameter, narrow-diameter, dental implants and mini dental implants, combined through advanced search in the aforementioned databases: (((“small”[All Fields] AND diameter[All Fields]) OR narrow-diameter[All Fields]) AND (“dental implants”[MeSH Terms] OR (“dental”[All Fields] AND “implants”[All Fields]) OR “dental implants”[All Fields])) AND (mini[All Fields] AND (“dental implants”[MeSH Terms] OR (“dental”[All Fields] AND “implants”[All Fields]) OR “dental implants”[All Fields])).

All articles were initially identified by two experts Georgina González-Valls (G.G.V.) and Juan Manuel Céspedes-Sánchez (J.M.C.S.). After reading the titles, the abstracts of those that presented the potential of inclusion were read. Subsequently, the full texts of the selected articles were read to verify that the inclusion criteria were met. Any discrepancies during the selection process were resolved by one of the two authors Beatriz González-Navarro (B.G.N.) or José López-López (J.L.L.).

### 2.5. Data Collection

The data were extracted by one of the authors Georgina González-Valls (G.G.V.) and entered in a data collection form (Microsoft Excel version 16.44). A second author reviewed the veracity of the collected data Elisabet Roca-Millan (E.R.M.). The following data were extracted: authors, type of study, number of patients, gender, age, diameter and length of the implants, location, type of restoration, follow-up time, number of failed implants, implants survival and success rates, and marginal bone loss.

### 2.6. Quality Assessment and Risk of Bias

Version 2 of Cochrane Collaboration’s tool for assessing the risk of bias in randomized trials (RoB 2) [15] was implemented to evaluate the risk of bias in the different domains of the included randomized clinical trials. The domains evaluated from the studies were: random sequence generation, allocation concealment, blinding of participants and researchers, blinding of outcome assessment, incomplete outcome data, selective outcome reporting and other sources of bias. Based on this, the studies were classified as ‘low risk’, unclear risk’, and ‘high risk’. 

The methodological index for non-randomized studies (MINORS) [16] was used to assess the risk of bias of the non-randomized clinical trials and cohort studies. In the MINORS scale, 12 categories were evaluated for comparative studies and 8 for studies that do not have a control group. 

### 2.7. Statistical Analysis

Different pooled estimates from the studies were obtained using the statistical analysis software OpenMeta [Analyst] (Version 1, Brown University, Providence, RI, USA).

Binary and continuous random-effects models were used to calculate the weighted proportions and means and the corresponding 95% confidence intervals (CI) among the studies. The evaluated variables were the survival, success, and failure rates and the marginal bone loss around dental implants, all at 36 months. Heterogeneity was assessed based on a calculation of the I^2^ statistic and the level of significance was set at *p* < 0.05.

## 3. Results

### 3.1. Study Selection

A total of 971 articles were identified in the first search phase. During the second phase and after reading the title of these, the systematic reviews, case reports, and expert opinions were discarded and those that dealt with mini-implants for orthodontic anchorage were also discarded; obtaining a total of 144 articles. Subsequently, the abstracts were read and 45 articles were obtained. Finally, after reading the full text, 15 articles were included in the review that met the established inclusion–exclusion criteria [17,18,19,20,21,22,23,24,25,26,27,28,29,30,31] (Figure 1).

### 3.2. Study Methods and Characteristics

Of the 15 articles included, 7 are clinical trials [17,19,20,21,22,27,31], 3 are randomized controlled clinical trials [18,24,28], 3 are cohort studies [25,26,29] and 2 are a case series [23,30], all of them published between 2010 and 2018 (Table 1).

The studies were carried out in 11 different countries: 2 in Belgium [17,24], 2 in Egypt [18,19], 1 in England [20], 2 in Italy [21,22], 1 in Switzerland [23], 1 in Turkey [25], 2 in Saudi Arabia [26,27], 1 in Brazil [28], 1 in China [29], 1 in the United States [30] and 1 in Spain [31].

The total number of patients included in the selected studies was 773 (359 men and 370 women). All papers included both men and women. The study by Al-Shibani et al. [27] was the only one in which gender was not specified. The study with the largest number of patients was Arisan et al. (N = 139) [25], while that of Moraguez et al. (N = 10) [23], was the one with the fewest patients. The mean age was 44.8 years, range from 13 to 80 years. In 5 articles [18,20,21,26,28] the age range was not specified and the article by Al-Aali et al. [26] was the only one in which the mean age was not specified.

The total number of implants was 1245. In all the studies except that of Galindo Moreno et al. [31] the area where they were placed was specified, with 713 implants placed in the mandible [17,18,19,20,21,22,24,25,26,27,28,29,30] and 433 in the maxilla [17,20,21,22,23,25,28,29,30]. All articles also specified whether the implants were placed anteriorly or posteriorly; with 998 implants placed in the anterior region [17,18,19,20,21,22,23,24,25,30,31] and 247 in the posterior region [26,27,28,29].

In all articles the diameter and length of the implants used are detailed. Four articles [17,18,19,30] used implants of less than 3.0 mm, four articles [20,21,22,31] between 3 and 3.25 mm, and seven articles [23,24,25,26,27,28,29] used implants between 3.3 and 3.5 mm. The most widely used diameters were those in the 3.3 to 3.5 mm category, with a total of 658 implants [23,24,25,26,27,28,29]. The next most used category was 3 to 3.25 mm with 306 implants [20,21,22,31] and lastly those of less than 3 mm with 281 implants [17,18,19,30] (Figure 2A). The mean length of the implants used was 12.3 mm (range 8 to 18 mm).

The main indication for NDIs in the reviewed studies was for anterior sector rehabilitation with single crowns [17,20,21,22,23,25,30,31]. A total of 533 implants were used in this area. The second most prevalent indication was rehabilitation with overdentures [18,19,24,25] with a total of 465 implants used and, finally, 247 implants were used in the rehabilitation of the posterior sector [26,27,28,29] (Figure 2B).

The mean follow-up time was 58 months from implant placement (range 36 to 137 months).

### 3.3. Quality Assessment and Risk of Bias

The present systematic review meets 24 of the 27 PRISMA criteria [14].

Table 2 shows the risk of bias of the three randomized clinical trials (RCTs) [18,24,28] according to the RoB 2 tool. Quirynen et al. [24] and de Souza et al. [28] have an overall low risk of bias. However, Quirynen et al. [24] does not specify the “Blinding of outcome assessment”. Maryod et al. [18] have an overall unclear risk of bias. Also it has a high risk of bias in “Blinding of outcome assessment” and an unclear risk of bias in “Blinding of participants and researchers”.

Of the 15 articles included in the present review, seven of them were NRCT [17,19,20,21,22,27,31] and three were cohorts [25,26,29], all of them evaluated according to the Minors scale (Table 3). Only one study [27] had a control group. The mean score obtained from non-comparative articles on the MINORS scale has been 11.8 points out of a total of 16, being Galindo-Moreno et al. [31] the article that has obtained the lowest score with 10 points. Lambert et al. [17], Elsyad et al. [19] and Al-Aali et al. [26] have been the articles that have obtained the maximum score of 13 points. Finally, Al-Shibani et al. [27] was the only comparative article. In this, a total of 12 items were evaluated, obtaining 15 points out of 24.

### 3.4. Marginal Bone Loss

Marginal bone loss was evaluated in all the articles except those by Lambert. et al. [17] and Maryod et al. [18]. Since most of the included studies had a follow-up period of 36 months or more, marginal bone loss was assessed up to this point in time. Six of the articles [19,22,23,24,28,31] could be included in the meta-analysis. A marginal bone loss of 0.821 mm was obtained (95% CI: 0.181–1.460; *p* = 0.012). Heterogeneity tests from pooled showed statistical significance (I^2^ = 99.451%, *p* < 0.001). Galindo Moreno et al. [31] obtained the least bone loss at 36 months with 0.110 mm. The diameter of the implants used was 3 mm. The study that recorded the greatest bone loss was that of Moraguez et al. [23] with 2.130 mm. The implants used had a diameter of 3.3 mm (Figure 3).

### 3.5. Failure, Success, and Survival Rates

Both the survival rate and the implant failure rate were described by all the studies except that of Al-Aali et al. [26]. However, the success rate was only evaluated in six of the 16 articles [19,22,24,25,28,30]. Meta-analyses were performed to assess these rates at 36 months. The survival rate was 97% (95% CI: 95.7%–98.3%; *p* < 0.001) with no significant difference regarding the heterogeneity of the included studies (I^2^ = 0%, *p* = 0.773) [18,19,20,21,22,24,27,28,31] (Figure 4). The failure rate obtained was 3% (95% CI: 1.7%n−4.3%; *p* < 0.001) without significant heterogeneity (I^2^ = 0%, *p* = 0.773 [18,19,20,21,22,24,27,28,31] (Figure 5). Only 4 articles evaluated the success rate at 36 months, with an estimated mean value of 96.8% (95% CI: 94% −99.6%; *p* < 0.001) and without statistically significant heterogeneity (I^2^ = 41.239 %, *p* = 0.164) [19,21,24,28] (Figure 6).

## 4. Discussion

NDIs are a valid treatment option that allows us to avoid more invasive procedures and higher morbidity. Patient preferences for minimally invasive treatment options, such as rehabilitation without bone augmentation, are generally high [32]. Therefore, this review aims to determine the survival and success rates of NDIs compared to SDIs.

Survival rates for NDIs are similar to those for SDIs. We place the survival rate of those between 96.7% and 99% [32]. Safii et al. [33] in 2010 stated the survival rate of SDI of 96.7% and Kim et al. [3] in 2018 of 98%. In the present review, most studies report figures for NDIs greater than 95% survival, and no study reports figures below 94.2%, with the mean survival rate at 36 months being 97% [18,19,20,21,22,24,27,28,31]. Reviewing the survival rates of implants associated with horizontal bone regeneration techniques, a survival of 96.2% [34,35] was obtained in the “Split crest technique” and of 98% when it was regenerated with autologous bone blocks or guided bone regeneration. These data suggest that treatment with NDI is a valid and safe therapeutic option that allows obtaining results similar to bone regeneration techniques.

When planning treatment with NDI, it is important to take into account factors such as its indications, the success of these, and the changes in bone remodeling that they may undergo, in addition to survival figures.

Most of the articles [19,21,25,30] take into account the criteria of Albrektsson et al. [36]: the implant is immobile when clinically evaluated, there is no evidence of radiolucency around it, the average vertical bone loss is less than 0.2mm per year, absence of pain, discomfort, or infection attributable to the implant, an implant design that allows the placement of a crown or prosthesis with a satisfactory appearance for both the patient and the dentist and a success rate of 85% at 5 years and 80% at 10 years [36]. One article [24] takes into account the criteria of Buser et al. [37], and another one [28] takes into account the criteria of Klein et al. [7] to determine the success of the implants. The success rate of the NDI described in our review is 96.8% at 3 years [18,28]. These results are similar to those obtained in implant placement procedures with simultaneous horizontal regeneration. Bassetti et al. [34] obtained a success rate of 88.2% to 100% and Waechter et al. [35] of 98.5%, both with the “Split crest technique”.

To date, there has been a lack of randomized clinical studies comparing the survival and success of NDI with that of SDI in regenerative procedures in situations of severe atrophy. Failed implants were those that presented significant mean bone loss, peri-implant radiolucency, mobility, infection, pain, discomfort, or that caused by sensorineural alteration. Forty-one implants out of a total of 1245 failed, the failure rate is 3% at 3 years [18,19,20,21,22,24,27,28,31]. Values very similar to those reported in other reviews [8,38]. The main reasons for failure were: infection [21,31], pain [18,25], and mobility [18,31]. Lemos et al. [39] stated in 2016 the failure rate of SDI of 2.72%.

Marginal bone loss over time is another important factor influencing the predictability of treatment. Assaf et al. [40], suggested that the predictability of the implant is not only related to its diameter but also the loss of marginal bone. These parameters should be within the same limits as those reported for SDIs. The acceptable bone loss established in the literature is 2 mm in the first year after loading the SDI, followed by a maximum of 0.2 mm per year [41,42]. Saffi et al. [33] reported the mean bone loss of SDI in 0.79 mm. It is noteworthy that the studies included in our review showed peri-implant bone loss comparable to that of SDIs. The mean bone loss reported in our review is 0.821 mm (CI: 0.181 to 1.460 mm) [19,22,23,24,28,31] at the end of the follow-up period (36 months), slightly higher than that of the SDIs, 0.695 mm at 3 years [43].

The NDIs were mostly used for restorations in the anterior sector, mainly upper lateral incisors or lower incisors. These locations usually have limited interdental space and a thin alveolar ridge. Placing an implant too close to an adjacent tooth can lead to interproximal bone loss, a factor that can negatively affect the final position of the papilla and the soft tissues at the supracrestal level [20]. Therefore, in the anterior sector, the aesthetics and stability of the peri-implant soft tissues are the main focuses of interest, in addition to the survival of the implant. King et al. [20] indicated that using NDI, soft tissue stability was achieved and clinically insignificant changes occurred in probing depth and gingival zenith level.

Overdentures as rehabilitation in edentulous jaws and with severe atrophy are another of the most frequent indications before the NDI. These are indicated in those cases in which we do not have enough bone to place an SDI and we do not want to perform any bone regeneration procedure. Therefore, we must be aware that NDIs are normally inserted in very atrophic edentulous jaws, which represents very challenging surgical situations [38]. Occlusion forces must also be considered. NDIs present limitations in certain clinical situations, such as in patients with bruxism [19].

Survival, success, and marginal peri-implant bone loss rates obtained in our review are similar to those of other reviews by other authors. The 97% survival obtained in our review is slightly higher than that obtained by other authors: 95.6% [7] and 96.5% [38]. The success obtained of 96.8% is higher than that obtained by Klein et al. [7] of 93.7% and comparable to that obtained by Schiegnitz et al. [38] of 96.2%. Regarding the peri-implant marginal bone loss obtained in our review (0.821 mm), it is comparable and slightly lower than that obtained by Schiegnitz et al. [38] of 0.993 mm and greater than that obtained by Klein et al. [7] of 0.53 mm.

The main limitation of this systematic review is the heterogeneity of the different articles included in terms of study design, follow-up period, sample size, and type of prosthetic rehabilitation. Some articles do not provide data on marginal bone loss, and others do not evaluate the success rate, or the data they provide is not sufficient to include them in the meta-analysis.

## 5. Conclusions

Based on the results obtained in this review, it can be stated that NDIs are a predictable treatment option. This is due to their favorable survival, success, and average bone loss rates. All of them are comparable to those of the SDI or those of regenerative bone treatments at a horizontal level and bone expansion techniques. Therefore, it can be concluded that NDIs are a valid therapeutic option in cases in which there is not enough bone volume in the horizontal direction to place an SDI. NDIs are a viable option in medically compromised or elderly patients to avoid more invasive procedures, thus reducing morbidity and treatment time.

## Figures and Tables

**Figure 1 materials-14-03234-f001:**
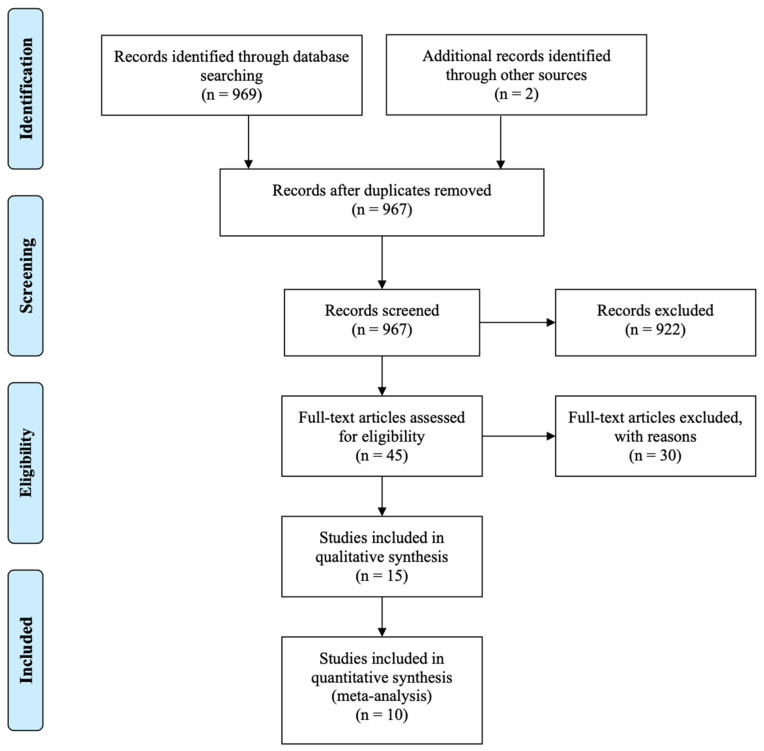
Preferred Reporting Items for Systematic Reviews and Meta-Analyses (PRISMA) flow diagram of selection process.

**Figure 2 materials-14-03234-f002:**
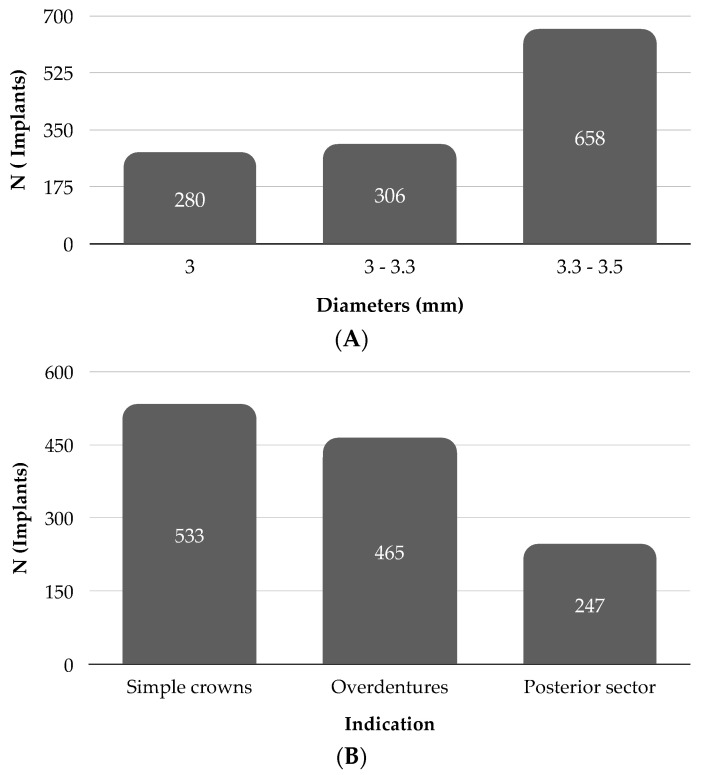
(**A**) Classification of implants according to diameter. (**B**) Classification of implants according to the indication.

**Figure 3 materials-14-03234-f003:**
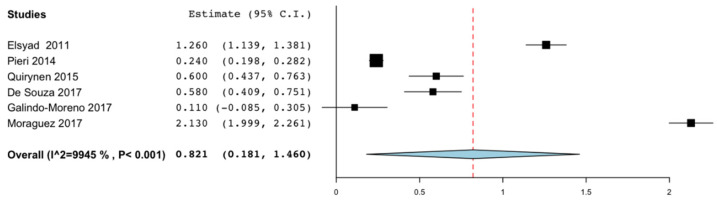
Peri-implant marginal bone loss at 36 months.

**Figure 4 materials-14-03234-f004:**
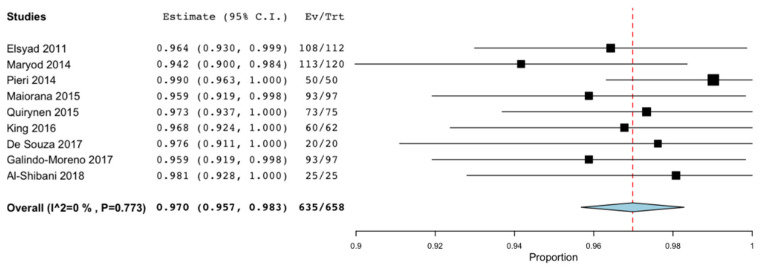
Survival rate at 36 months.

**Figure 5 materials-14-03234-f005:**
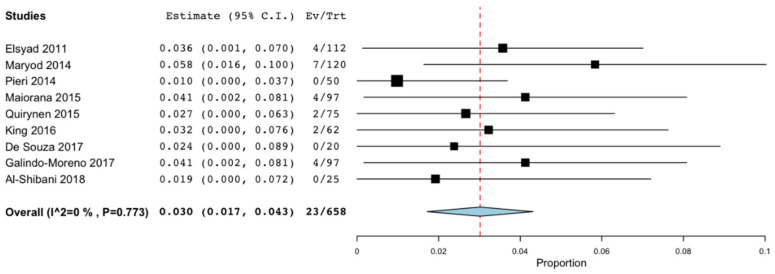
Failure rate at 36 months.

**Figure 6 materials-14-03234-f006:**
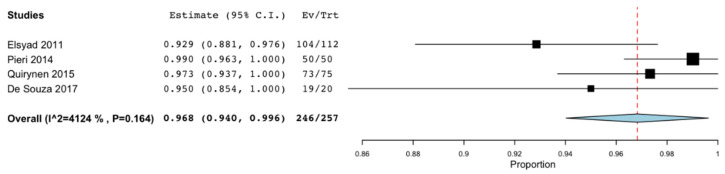
Success rate at 36 months.

**Table 1 materials-14-03234-t001:** Characteristics of the articles included. M: Males; F: Females; MAX: Maxilla; MAND: Mandible; RCT: Randomized Clinical Trial; CT: Clinical Trial; CS: Case Series; ND: No data.

Author	Type of Study	Number of Patients		Age (Range)	Diameter (mm) Length (mm)	Number of Implants	Location (Maxilla/ Mandible)	Type of Restoration	Follow-up	Failure; Survival Rate; Succes Rate, Implant	Mean Marginal Bone Loss (mm)
		M	F								
Lambert et al. 2017 (17)	CT	6	14	15.6 (13–19)	2 × 10; 13	30	MAX/MAND	Single crowns in anterior sector	42	1; 96.6%; ND	−1.09 ± 1.74
					2.5 × 10; 13						
Maryod et al. 2014 (18)	RCT	20	16	64.1	1.8 × 15	120	MAND	Overdenture	36	7; 94.2%; ND	ND
Elsyad, et al. 2011 (19)	CT	16	12	62.9 (49–75)	1.8 × 12; 18	112	MAND	Overdenture	36	4; 96.4%; 92.90%	−1.26 ± 0.64
King et al. 2016 (20)	CT	18	20	24	3.0 × 11; 15	62	MAX/MAND	Single crowns in anterior sector	36	2; 96.8%; ND	−0.23
Maiorana et al. 2015 (21)	CT	36	33	32	3.0 × 11; 15	97	MAX/MAND	Single crowns in anterior sector	36	4; 95.9%; ND	−0.09
Pieri et al. 2014 (22)	CT	18	32	41.58 (19–64)	3.0 × 11; 15	50	MAX/MAND	Single crowns in anterior sector	36	0; 100%; 100%	−0.24 ± 0.15
Moraguez et al. 2017 (23)	CS	4	6	49.4 (32–68)	3.3 × 10; 12	20	MAX	Single crowns in anterior sector	60	0;100%; ND	−2.17 ± 0.38
Quirynen et al. 2015 (24)	RTC	40	49	65.8	3.3 × 8; 14	75 (titanium implants)	MAND	Overdenture	36	2; 97.3%; 97.3%	−0.6 ± 0.71
Arisan et al. 2010 (25)	Cohort	66	73	55.3 (21–80)	3.3 × 8; 14, 9.5; 15	316	ND	Overdenture; Single crowns in anterior sector	124	14; 95.6%; 91.40%	−1.32 ± 0.13 maxilla
					3.4 × 8; 14, 9.5; 15						−1.28 ± 0.14 mandible
Al-Aali et al. 2018 (26)	Cohort	43	35	≥25	3.3 × 10; 12	102	MAND	Crowns posterior sector	42	ND	−1.17 ± 0.06
Al-Shibani et al. 2018 (27)	CT	44		41.6 (30–50)	3.3 × 10	25	MAND	Crowns posterior sector	36	0; 100%; ND	0.15 (0.1–0.4)
De Souza et al. 2017 (28)	RTC	10	12	59.2	3.3 × 6; 12	22	MAX/MAND	Crowns posterior sector	36	0; 100%; 95%	−0.58 ± 0.39
Yu-Shi et al. 2017 (29)	Cohort	38	29	35.6 (21–56)	3.3 × 10; 12	98	MAX/MAND	Crowns posterior sector	121	3; 96.9%; ND	−1.19 ± 1.07
Froum et al. 2017 (30)	CS	6	8	48.6 (23–87)	1.8 × 7; 10; 14	19	MAX/MAND	Single crowns in anterior sector	137	0; 100%; 84%	−0.16
					2.2 × 7; 10; 14						
					2.4 × 7; 10; 14						
Galindo-Moreno et al. 2017 (31)	CT	36	33	32.5 (18–72)	3 × 11; 13; 15	97	MAX/MAND	Single crowns in anterior sector	60	4; 92.8%; ND	−0.15

**Table 2 materials-14-03234-t002:** Risk of Bias of included randomized clinical trials. +: Low; ?: Unclear; −: High.

Author	Random Sequence Generation	Allocation Concealment	Blinding of Participants and Researchers	Blinding of Outcome Assessment	Incomplete Outcome Data	Selective Outcome Reporting	Other Sources of Bias
Maryod et al. 2014 (18)	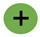	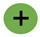	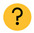	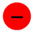	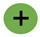	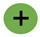	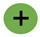
Quirynen et al. 2015 (24)	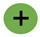	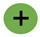	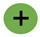	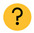	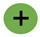	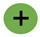	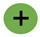
De Souza et al. 2017 (28)	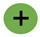	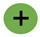	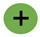	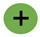	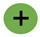	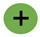	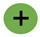

**Table 3 materials-14-03234-t003:** Risk of bias of included non-randomized clinical trials. 0 (not reported), 1 (reported but inadequate) or 2 (reported and adequate), NE (not evaluable, no group control).

Author	A Clearly Stated Aim	Inclusion of Consecutive Patients	Prospective Collection of Data	Endpoints Appropriate to the Aim of the Study	Unbiased Assessment of the Study Endpoint	Follow-up Period Appropriate to the Aim of the Study	Loss to Follow up Less than 5%	Prospective Calculation of the Study Size	An Adequate Control Group	Contemporary Groups	Baseline Equivalence of Groups	Adequate Statistical Analyses
Lambert et al. 2017 (17)	2	2	2	2	1	2	2	0	NE	NE	NE	NE
Elsyad, et al. 2011 (19)	2	2	2	2	0	2	1	2	NE	NE	NE	NE
King et al. 2016 (20)	2	2	2	2	0	2	2	0	NE	NE	NE	NE
Maiorana et al. 2015 (21)	2	2	2	2	0	2	2	0	NE	NE	NE	NE
Pieri et al. 2014 (22)	2	2	2	2	0	2	2	0	NE	NE	NE	NE
Arisan et al. 2010 (25)	2	2	2	1	0	2	2	0	NE	NE	NE	NE
Al-Aali et al. 2018 (26)	2	2	2	2	0	2	2	1	NE	NE	NE	NE
Al-Shibani et al. 2018 (27)	2	2	2	2	0	2	1	0	0	2	2	0
Yu-Shi et al. 2017 (29)	2	2	2	2	0	2	1	0	NE	NE	NE	NE
Galindo-Moreno et al. 2017 (31)	2	2	2	2	0	2	0	0	NE	NE	NE	NE

## Data Availability

Data are contained within the article.

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
