# Peer review of "Narrow Diameter Dental Implants as an Alternative Treatment for Atrophic Alveolar Ridges. Systematic Review and Meta-Analysis"

_materials, 2021, doi:10.3390/ma14123234_

Round 1

Reviewer 1 Report

Executive Summary

The manuscript titled “Narrow Diameter Dental Implants as an Alternative Treatment for Atrophic Alveolar Ridges. Systematic Review and Meta-Analysis” searched literature regarding NDIs and their treatment results. Further, the authors performed a meta-analysis to decide the effectiveness and safety of these treatments. Overall, the manuscript is well written. Due to some technical problems, the authors did not list Table 2 and Table 3 in the manuscript. Therefore, I could not finish the review due to the missing information. The authors may perform major revisions.

Major Comments

  • Table 2 shows the risk of bias of the 3 RCTs [17,23,27] according to the RoB2 tool.
    • Please insert Table 2 in the designated location.
  • Of the 15 articles included in the present review, 7 of them were NRCT [17,19-22,27,31] and 3 were Cohorts [25,26,29], all of them evaluated according to the Minors scale (Table 3).
    • Please insert Table 3 in the designated location.

Minor Comments

  • For Table 1, please provide full terms for all the shortcuts as footnotes, including but not limited to CT, RCT, MAX, MAND.

Reviewer 2 Report

This systematic review and meta-analysis focused on narrow diameter implant (NDI), which are useful for thin and limited bone to avoid bone augmentation surgery. The information was interesting and beneficial for readers, but some revisions were necessary.

Introduction

In 4th paragraph, “small diameter dental implants (NDI)”  This description should include “narrow” because the abbreviation is “NDI”.

Results

The structure should be revised. Blank page was found.

All number of references should be parenthesized.

Discussion

The results of SDI were not clear. The aim of this study was to compare NDI with SDI, so some references should be described clearly in Discussion, too.

The authors mentioned the success criteria by Albrektsson et al. The success criteria might be different in cited references. The author should described detailed criteria clearly if you found different criteria in the references.

Reviewer 3 Report

Authors performed an extensive study: "Narrow Diameter Dental Implants as an Alternative Treatment 2 for Atrophic Alveolar Ridges. Systematic Review and Meta-3 Analysis"
Meta-analyzes were performed to assess 20 marginal bone loss and implant survival, success, and failure rates. A total of 971 articles were analyzed, and finally 15 studies were included: 7 clinical trials, 3 randomized clinical trials, 3 cohort studies, and 2 case series.

Introduction is well written and in my opinion, there is no need for changes.

Page 2, 2.3. Eligibility Criteria
In Exclusion criteria: i) Systematic re-85 views, case reports, and expert opinions; ii) Animal studies; iii) Studies in which the sam-86 ple was less than 10 subjects; iv) Follow-up period of less than 36 months after implant 87 placement; v) Smokers of more than 10 cigarettes/day; vi) Mini-implants for orthodontic 88 anchoring.
What was the reasoning behind the Exclusion criteria? Why were the studies with less than 10 subjects rejected? There is one study with exactly 10 subject, and I am wandering is there any particular reason for 10 or it is an arbitrary measure? Because, at the end these exclusion criteria resulted with only 15 studies.

Page 5, a bar graph for investigated diameters would be very helpful here.

Page 9, "0.110mm. The diameter of the implants used was 3mm. The study 208 that recorded the greatest bone loss was that of Moraguez et al.23 with 2,130mm. The 209 implants used had a diameter of 3.3mm (Figure 2)."
Sometimes a decimal point is used and sometimes a comma. This should be standardized.

Page 9, Figure 2.
X-axis should be extended because the last point is outside of the graph. Which values are on the y-axis?

Page 11.
What is the meaning of "Bassetti et al. [34] obtained a suc-255 cess rate of 88.2 to 100%" success rate of 88.2 to 100%?

Page 12.
Authors state: "Survival, success, and marginal peri-implant bone loss rates obtained in our review are similar to those of other reviews by other authors."
Is there any reason why the authors should obtain different survival, success, and marginal peri-implant bone loss rates?

Round 2

Reviewer 1 Report

The revised version of the manuscript titled “Narrow diameter dental implants as an alternative treatment for atrophic alveolar ridges. Systematic review and meta analysis” has addressed reviewers’ comments and has met the requirement for publication. The following suggestion is purely for the authors’ information only. The authors may or may not need to perform any revisions.

Table 3: Since only one reference provides information about An adequate control group, Contemporary groups, Baseline equivalence of groups, and Adequate statistical analyses, the authors may consider remove these four columns and use “*” and table footnote to provide such information. The goal is to improve reader friendliness.

Reviewer 2 Report

I’ve checked a revised one, but it was insufficient for acceptance. As I mentioned in the first comment, the authors should describe the results of SDI using the previous studies. I found it as reference #6, but it was published in 1997. In addition, the authors should describe clearly that whether their purpose was to compare the results of NDI with SDI placed in narrow bone with some technique or normal bone for SDI.

And the tables should be adjusted. In Table 3, the blanks were found and these were inappropriate. If you found nothing, you should add “0”.

Round 3

Reviewer 2 Report

Get improved.